# Advances in Diagnostics and Drug Discovery against Resistant and Latent Tuberculosis Infection

**DOI:** 10.3390/pharmaceutics15102409

**Published:** 2023-09-30

**Authors:** Christian Shleider Carnero Canales, Jessica Marquez Cazorla, André Henrique Furtado Torres, Eloise T. Monteiro Filardi, Leonardo Delello Di Filippo, Paulo Inácio Costa, Cesar Augusto Roque-Borda, Fernando Rogério Pavan

**Affiliations:** 1Facultad de Ciencias Farmacéuticas Bioquímicas y Biotecnológicas, Vicerrectorado de Investigación, Universidad Católica de Santa María, Arequipa 04000, Peru; christian.carnero@unesp.br (C.S.C.C.);; 2Institute of Chemistry, São Paulo State University (UNESP), Araraquara 14801-970, SP, Brazil; 3School of Pharmaceutical Sciences, São Paulo State University (UNESP), Araraquara 14801-970, SP, Brazil; 4Department of Drug Design and Pharmacology, Faculty of Health and Medical Sciences, University of Copenhagen, 2300 Copenhagen, Denmark

**Keywords:** diagnostic, Drug discovery, *Mycobacterium tuberculosis*, Latent tuberculosis infection

## Abstract

Latent tuberculosis infection (LTBI) represents a subclinical, asymptomatic mycobacterial state affecting approximately 25% of the global population. The substantial prevalence of LTBI, combined with the risk of progressing to active tuberculosis, underscores its central role in the increasing incidence of tuberculosis (TB). Accurate identification and timely treatment are vital to contain and reduce the spread of the disease, forming a critical component of the global strategy known as “End TB.” This review aims to examine and highlight the most recent scientific evidence related to new diagnostic approaches and emerging therapeutic treatments for LTBI. While prevalent diagnostic methods include the tuberculin skin test (TST) and interferon gamma release assay (IGRA), WHO’s approval of two specific IGRAs for *Mycobacterium tuberculosis* (MTB) marked a significant advancement. However, the need for a specific test with global application viability has propelled research into diagnostic tests based on molecular diagnostics, pulmonary immunity, epigenetics, metabolomics, and a current focus on next-generation MTB antigen-based skin test (TBST). It is within these emerging methods that the potential for accurate distinction between LTBI and active TB has been demonstrated. Therapeutically, in addition to traditional first-line therapies, anti-LTBI drugs, anti-resistant TB drugs, and innovative candidates in preclinical and clinical stages are being explored. Although the advancements are promising, it is crucial to recognize that further research and clinical evidence are needed to solidify the effectiveness and safety of these new approaches, in addition to ensuring access to new drugs and diagnostic methods across all health centers. The fight against TB is evolving with the development of more precise diagnostic tools that differentiate the various stages of the infection and with more effective and targeted treatments. Once consolidated, current advancements have the potential to transform the prevention and treatment landscape of TB, reinforcing the global mission to eradicate this disease.

## 1. Introduction

Tuberculosis (TB) is caused mainly by *Mycobacterium tuberculosis* (MTB), an actinomycete closely related to saprophytic bacteria such as *Mycobacterium smegmatis*. Unlike most Gram-positive bacteria, MTB bacilli increase their peptidoglycan wall with a wide range of complex lipidoglycans, and this characteristic has posed significant challenges for the treatment and eradication of the disease. Despite continuous efforts to control and prevent TB, it remains a significant burden on global public health [1].

According to the 2022 report from the World Health Organization [2], 25% of the global population is infected with a latent form of tuberculosis (LTBI). This carries a lifetime risk of progressing to active disease ranging between 5% and 10%. The report also highlights that the primary focus in terms of diagnosis and treatment is currently on children and young adolescents (under 15 years of age), who account for approximately 11% of the global burden. This equates to an annual incidence of 1.1 million children contracting the disease. It is particularly alarming that nearly half of these are under five years of age, a demographic in which the mortality rate approaches 80%. Furthermore, adolescents aged 15 to 19 years represent another significant segment of infection, with over half a million active cases reported each year.

TB is primarily transmitted through microdroplets expelled by actively infected individuals when coughing, sneezing, or talking. These droplets can remain suspended in the air and infect nearby individuals [3]. The first line of defense against MTB is the respiratory mucosa, which contains a range of protective molecules such as immunoglobulins A, antimicrobial peptides (AMPs), antibodies, cytokines, and chemokines. When these molecules fail to control the invasion in the first instance, MTB disseminates to the pulmonary alveoli, and shortly thereafter alveolar macrophages, dendritic cells, and neutrophils are activated to attempt to eradicate the infection [4]. MTB has evolved and developed the ability to internalize macrophages to live within phagosomes, thereby overcoming nutrient availability and stressful environmental factors, while providing protection against drugs and therapeutic doses (Figure 1) [5].

Among the people infected with MTB, there is a group of patients who develop an inactive form of TB (latent TB). MTB remains inactive, causing recurrent or chronic infections depending on the host’s immunological conditions [5,6]. Macrophages generally produce AMPs, cytokines, reactive oxygen species (ROS), and nitric oxide (NO) to eliminate pathogens [7]. However, MTB can inhibit these intracellular immune response mechanisms by promoting the production of IL-10. Prolonged exposure to this cytokine also affects the apoptosis, phagocytosis, autophagy, and Th1 immune response of myeloid cells [8]. All these functions are mediated by IL-10, as it binds to its receptor to activate STAT3, a protein that stimulates the transcription of specific genes that transcriptionally repress proinflammatory cytokine genes [9]. Additionally, proinflammatory cytokines are important for the retention of lymphocytes in non-lymphoid sites after MTB infection [10].

Latent tuberculosis infection (LTBI) is defined by the World Health Organization [11] as a “state of persistent immune response to stimulation by MTB antigens with no evidence of clinically manifest active TB”. It is triggered by a sustained immune response against MTB antigens and self-healing lesions that form granulomas, which act as a type of bacterial reservoir. Patients with LTBI are asymptomatic; they are infected with MTB but do not develop active TB disease. Furthermore, patients with LTBI are not infectious and cannot transmit TB infection to others [12]. Although TB is not a new disease, incidence rates are still high, and concern is growing due to the lack of a specific treatment to combat LTBI, which is difficult to treat due to the disease’s physiopathological characteristics and its challenging diagnosis [13].

This review aims to examine and highlight the latest scientific evidence regarding new diagnostic approaches and emerging therapeutic treatments for LTBI.

## 2. Epidemiology of Latent Tuberculosis Infection

Roughly two decades ago, it was estimated that a third of the global population had LTBI. However, demographic changes, advancements in pharmacology, and enhanced TB control strategies have significantly revised this estimate. Currently, it is believed that 25% of the global population carries this latent infection [14]. The importance of controlling and preventing LTBI lies in its worldwide distribution and its high mortality rate when progressing to active disease, which caused 1.3 million recorded deaths in 2020, nearly twice the number of deaths caused by HIV [15]. Houben and Dodd [16] used mathematical models to estimate the global incidence of LTBI. Their findings indicated that, in 2014, 23% of the global population (approximately 1.7 billion people) was infected, with Southeast Asia, the Western Pacific, and Africa being the regions with the highest prevalence, representing approximately 80% of the global burden.

These findings are in agreement with the most recent update by Cohen et al. [17] on the prevalence of LTBI burden in a meta-analysis study based on the results of the interferon-γ release assays (IGRAs) and the tuberculin skin test (TST). This study reported a global prevalence of LTBI of 24.8% (95% confidence interval [CI 95%]: 19.7–29.9%) and 21.2% (CI 95%: 17.9–24.4%) for IGRAs and TST, respectively. However, it is important to emphasize that obtaining an accurate estimate of the LTBI rate remains a significant challenge due to the lack of a uniform standard for identifying this condition. China and India are the countries with the highest prevalence of LTBI, with approximately 350 million infected individuals in China [18] and an infection rate in India ranging between 40% and 50% in various populations, implying that more than 500 million inhabitants may have LTBI [19]. In the United States, it is estimated that approximately 13 million people have LTBI [20], with 80% of TB cases resulting from untreated latent infection [21]. In Africa, infection rates vary between 31.2% in Ethiopia, 49% in Uganda, and 55.2% in South Africa, with high prevalence in at-risk populations such as miners (89%) and healthcare workers (62–84%) in high-incidence areas. Between 5% and 15% of individuals with LTBI progress to active TB, with a higher risk associated with impaired immunity [22].

In the WHO European Region, the prevalence of LTBI is estimated at 13.7%, with 0.3% being recent infections at a higher risk of progressing to active TB. In low-incidence countries, most TB cases are generated through reactivation of LTBI acquired abroad due to global migration (Figure 2). Therefore, a significant challenge for European control programs is to establish a programmatic approach to LTBI management [23]. Despite the decrease in the global burden of LTBI, declining from one third to one quarter of the world’s infected population, significant challenges persist. Until 2015, there were 19.1 million people reported to be latently infected with high-risk multidrug-resistant TB (MDR-TB). MDR strains accounted for 1.2% of the overall LTBI burden and 2.9% of the burden among children under 15 years of age [24]. Currently, these numbers are increasing due to complications in lung diseases caused by the COVID-19 pandemic, posing serious challenges for LTBI management, a cornerstone of TB elimination strategies. Concerted efforts must be made globally to address the prevention and treatment of LTBI, with an emphasis on high-prevalence areas and at-risk populations, in order to achieve WHO’s objectives and reduce the global burden of this devastating disease [25].

## 3. New Advances in The Diagnosis of LTBI

Two methods have been described as the most commonly used in the diagnosis of TB and probable screening tests for LTBI. One diagnostic method is the tuberculin skin test (TST), which is based on a purified protein derivative (PPD) of proteins secreted by MTB. This solution is injected intradermally into the patient’s forearm, aiming to induce a delayed hypersensitivity reaction caused by the T lymphocytes that will generate local inflammation mediated by various cytokines within a period of 48 to 72 h [26,27,28]. However, a limitation of the clinical application of this test is that the TST contains a mixture of non-specific antigens for MTB, which can result in false-positive results in patients previously vaccinated with Bacillus Calmette-Guerin (BCG) or patients who have been infected by non-tuberculous mycobacteria. On the other hand, false-negative results may occur in immunocompromised patients with insufficient T cell responses to generate the local hypersensitivity reaction [29,30,31].

In the quest for better specific tests that can more accurately identify the development of active TB in patients, interferon gamma release assays (IGRAs) have been developed. This in vitro blood analysis is based on T-cell-mediated immunity to specific antigens of MTB. The test is considered sensitive and specific for active TB and LTBI, as it was developed from two MTB-specific T-cell antigens: ESAT-6 and CFP-10 [32,33]. However, like the TST, this test is limited by its inability to differentiate active TB from LTBI. As a result, alternative detection tests are being sought or combinations of these. In the context of LTBI diagnosis in patients with inflammatory bowel disease undergoing immunosuppressive therapy (IST), Park et al. [34] suggest that the optimal diagnostic strategy is the combination of IGRA followed by TST. This sequence significantly improves accuracy in detecting LTBI, particularly in situations after IST initiation and in populations previously vaccinated with BCG. On the other hand, sole reliance on IGRA, especially in patients under IST, represents a less reliable approach; even with a negative IGRA result, there remains an underlying risk of tuberculous reactivation. This can be attributed to the ability of IST to decrease the positivity rate of IGRA. Additionally, the variability in TST efficacy, depending on the chosen cutoff value, and the potential drop-in positivity rates with fixed cutoffs during IST, underscore the necessity for a combined approach. It is worth noting that there have been documented cases of progression to active tuberculosis despite a negative IGRA outcome, emphasizing the significance of a thorough diagnostic strategy. On the other hand, Kang et al. [35] report that the combination of IGRAs and complete blood count (CBC) analysis was investigated as a possible method for differentiating active TB from LTBI. A total of 126 samples (blood, serum, and plasma) were used, and the authors concluded that the statistically most significant blood biomarkers in comparing their expression levels were total white blood cells, neutrophils, lymphocytes, and monocytes. However, further studies with a larger number of clinical samples are needed to confirm the importance of these biomarkers and improve the accuracy in differentiating MTB pathological states.

The pace of development of more accurate tests for LTBI has been slow. In 2001, the FDA approved a first-generation test, the QuantiFERON-TB. Subsequently, in 2005, a second generation of this test, known as QuantiFERON-TB GOLD in Tube (QFT), was approved. In 2008, another test, the T-SPOT.TB, was also approved. These tests are based on the IGRA. Today, these IGRAs are commercially available and are endorsed by Health Canada. Additionally, they carry the CE (Conformité Européenne) mark, certifying their use in Europe, and have received the exclusive endorsement of the World Health Organization for their implementation [28,36].

The QFT assay is an enzyme-linked immunosorbent assay (ELISA) that uses peptides from the RD1 ESAT-6 and CFP-10 antigens, targeting CD4 T-helper cell-mediated immune responses in a tube. Results are reported as quantification of IFN-γ [37]. There is currently a new generation of IGRA available, specifically the QuantiFERON-TB Gold Plus (QFT-Plus) assay. The difference with its predecessor lies in the addition of a new antigen specific to CD8 T cells in a second antigen tube (QFT-Plus Tube 2, TB2) specifically designed to stimulate CD8 and CD4 T cells. In this way, the new antigen tube complements the first antigen tube (QFT-Plus Tube 1, TB1) [38]. The importance of CD8 T cells in TB infection lies in their participation in the recognition and destructive attack on cells infected with MTB. They also have an effect during the replication of MTB in its active phase, which decreases during the treatment phase. Based on this evidence, a higher frequency of CD8 T cells is found in patients with active TB than in LTBI patients, indicating a direct relationship with antigenic load [39].

In 2018, Petruccioli et al. [40] evaluated the response to TB treatment in relation to the results of the QFT-Plus test. They contacted 74 participants in different stages of TB infection, including 46 participants with LTBI and 28 patients with active TB. The study showed that therapies can reduce the level of IFN-γ in response to stimulation by TB1 and TB2 peptides. However, this result varied greatly depending on various factors, the most impactful of which occurred in patients with clinical diagnosis, in comparison with those with microbiological diagnosis. This difference is likely due to the bacterial load maintaining a latent immune response, which is why the QFT-Plus test cannot yet be considered a definitive tool for differentiation and monitoring in the treatment of LTBI and active TB. On the other hand, the T-SPOT.TB test is an enzyme-linked immunospot assay (ELISPOT) performed on isolated and counted peripheral blood mononuclear cells (PBMCs) incubated with ESAT-6 and CFP-10 peptides. Results are reported as the number of IFN-γ-producing T cells (spot-forming cells). Inconclusive IGRA results may occur due to a low IFN-γ response to positive controls (mitogens) or a high background response to negative controls [36].

In order to compare the agreement and effectiveness of the two IGRAs (QFT-GIT and T-SPOT.TB) in the diagnosis of active TB, Du et al. [41] conducted a comparative study and concluded that both the sensitivity and specificity of T-SPOT.TB were slightly higher than those of QFT-GIT but did not reach statistical significance. This is likely due to the difference in the more complex technical characteristics involved in this test. When different groups were investigated, the concordances were 93.4%, 90.0%, and 93.7% in the confirmed TB, probable TB, and non-TB groups, respectively. Despite these results, no test is entirely suitable for the diagnosis of LTBI due to sensitivity and specificity issues, inability to distinguish infections with MDR-MTB strains, and the high costs involved.

In this line of research, concerning the development and optimization of different assays for the diagnosis and differentiation of TB and LTBI, surface markers on T cells were evaluated as a starting point, since these are the cells responsible for generating an immune response against MTB infection. Yang et al. [42] investigated the clinical utility of T cells expressing the CD161 marker in the effective differentiation of active TB. For this research, eight markers were selected using flow cytometry, with CD161 standing out as the promising marker for the differentiation of active TB from LTBI. Among the results obtained, it was observed that the percentage of T cells expressing CD161 was lower in active TB than in LTBI or healthy controls. However, these data were not sufficient, which is why the lymphocyte/monocyte ratio was included as a differentiation parameter, as these are also significantly reduced in an active TB infection. Although it was concluded that the CD161 marker measurement method has high specificity and sensitivity for differentiating active TB from LTBI, its main limitation in clinical application is that not all hospitals have flow cytometers.

As shown, most diagnostic tests are based on the immune response generated in peripheral blood. However, Das et al. [43] reported that MTB infection primarily targets the lung, highlighting the importance of exploring the pulmonary immune response, including alveolar T cells and alveolar macrophages. In addition, it has recently been demonstrated that DNA methylation is fundamental for the immune response, as confirmed by Karlsson et al. [44] who compared and evaluated epigenetic modifications in DNA methylomes in both alveolar macrophages and alveolar T cells. They identified a particular and different DNA methylation profile in patients without peripheral immune response who developed LTBI during the study. The differentially methylated genes (DMGs) identified in subjects who developed LTBI were overrepresented in the pentose phosphate pathway of alveolar macrophages and in IFN-γ signaling and migration in alveolar T cells. Finally, they argued that knowing the DNA methylation status in pulmonary immune cells can indicate who will develop LTBI. Another significant finding was that the identification of this methylation profile detects an early result of an LTBI infection, before any other available diagnostic method.

Similarly, Liu et al. [45] developed Genepop, a simple, rapid, and low-cost diagnostic method characterized by not requiring a cold chain, with a concordance of 91.6% based on loop-mediated isothermal amplification. This test is useful in domestic or field testing, and its importance lies in helping to prevent the spread of MTB in several countries due to its low production cost and easy operability. Despite the advances in developing new diagnostic methods for LTBI, MDR-TB strains have hindered the development of an effective and specific method for accurately diagnosing LTBI.

In pediatric populations, the challenge of diagnosing TB is notably accentuated. Traditional bacteriological methods often fall short due to the limited bacillary loads in children, leading to frequent misdiagnoses or confusions with other pediatric conditions. This diagnostic shortfall has contributed to a rising incidence and mortality from TB among the young. In light of this, starting in 2021, the World Health Organization has championed the integration of cutting-edge molecular diagnostic techniques (Table 1). These are designed to either supersede or complement existing methods, thereby amplifying the sensitivity and specificity of TB diagnosis in both children and adults [2,46].

In 2021, the World Health Organization [46] issued guidelines advocating for tubercular immunological MTB antigen-based skin test, termed TBST. These assays emphasize the ESAT-6 protein and filtrate protein 10 (CFP-10) antigens to trigger the release of IFN-γ from specific T cells. Among the TBSTs evaluated by the WHO are: Cy-Tb (India), which integrates recombinant proteins derived from modified *Lactobacillus lactis* genes, with a 0.1 mL dose containing 0.05 μg of rd ESAT-6 and 0.05 μg of rCFP-10; Diaskintest^®^, formulated with a recombinant protein produced by *E. coli* BL21 (DE3)/pCFP-ESAT, with a 0.1 mL dose delivering 0.2 μg of the recombinant ESAT-6 and CFP10 proteins, endorsed by Russian authorities; and C-TST, employing a recombinant fusion protein of ESAT-6 and CFP-10, expressed in genetically optimized *E. coli*, with a 0.1 mL dose providing five units (U) of the recombinant Mtb fusion protein, sanctioned by the Chinese government.

Central to the innovation of these tests is their fusion of IGRA specificity with the skin test platform. A study by Hamada et al. [47] juxtaposed the diagnostic capabilities of Diaskintest, ELISPOT, and QFT, revealing sensitivities of 88.7%, 90.6%, and 87.0%, respectively. This underscores the comparable specificity of TBSTs to IGRAs, surpassing that of the TST. Moreover, their specificity matches or even exceeds that of IGRA in pediatric populations and those co-infected with HIV. This was further corroborated by Nikitina et al. [48] who compared Diaskintest and QuantiFERON-TB Gold (QFT) in adults and children suspected of having TB. Their findings indicated an 84% concordance in adults and 90% in children, with heightened diagnostic sensitivity in the latter (73% and 65%, respectively). However, this evidence has yet to undergo systematic review.

In addition to diagnostic sensitivity validation, Starshinova et al. [49] demonstrated that TBSTs are as safe to use as the TST, with over 95% of subjects experiencing only mild injection site reactions, such as swelling, slight pain, and itching. Given its simplicity and cost effectiveness, the TBST emerges as a promising tool to enhance detection and health equity, mitigating the risk of false positives.

Concurrently, there is an ongoing pursuit for novel biomarkers, beyond DNA, that hold the potential to refine and advance more efficient diagnostic methodologies [50]. In this context, there has been increasing interest in the application of untargeted metabolomics, a tool deemed strategic for the identification of a wide range of significant biomarkers, thereby facilitating a better understanding of the host–pathogen interaction [51].

The research conducted by Conde et al. [52] focused on uncovering a set of biomarkers associated with TB diagnosis by conducting a metabolic study of serum through nuclear magnetic resonance. The study group included healthy individuals, TB patients (active TB or LTBI), and patients with pulmonary and extrapulmonary TB. The study indicated that the metabolism of amino acids was the most significantly altered and differentiated, followed by purine metabolism, glyoxylate and dicarboxylate metabolism, and aminoacyl-tRNA biosynthesis. They concluded that inosine, hypoxanthine, mannose, asparagine, aspartate, and glutamate were the six primary metabolites prominently linked to metabolic processes during TB infection. These results can be interpreted considering that the MTB bacterium thrives inside macrophages, confronting an acidic environment with nutrient scarcity.

Metabolomics is also useful for differentiating between various species and TB-XDR strains selectively. In line with this, Huang et al. [53] identified TB-XDR biomarkers using ultra-high-performance liquid chromatography in conjunction with mass spectroscopy (UPLC-Q-TOF-MS), pinpointing four primarily altered metabolites: N1M2P5C, MG3P, CA, and DX. With these, they were able to construct a differential diagnostic model for TB-XDR with an accuracy, sensitivity, and specificity of 0.928, 86.7%, and 86.7%, respectively. Conversely, Chaiyachat et al. [54] discovered metabolic markers that differentiate pre-XDR strains from MTB-XDR using ultra-high-performance liquid chromatography alongside electrospray ionization quadrupole time-of-flight mass spectrometry (UHPLC-ESI-QTOF-MS/MS). They reported that the levels of the metabolites meso-hydroxyheme and itaconic anhydride are responsible for distinguishing between pre-XDR and XDR strains.

Metabolomics can also contribute to the development of methods to differentiate between TB and LTBI. Albors-Vaquer et al. [55] examined the differences in the metabolic profile of patients with active TB and those with LTBI, showing that both groups exhibit a unique serum metabolic profile compared with healthy individuals. They reported that the levels of amino acids such as alanine, lysine, glutamate, glutamine, citrate, and choline decrease in patients with an active infection, as MTB quickly absorbs and metabolizes them as nitrogen sources. In line with this finding, Della Bella et al. [56] developed a new blood analysis called LIOSpot TB that distinguishes TB from LTBI. This test is based on alanine dehydrogenase and has demonstrated that the MTB antigen can stimulate the production of IL-2 in active TB but not in LTBI. Although further validation and comparison with other pulmonary diseases for clinical evaluation are still necessary, these findings represent a significant advancement, highlighting the potential of metabolomics to provide new study targets.

## 4. Current Treatment Regimen

The conventional treatment for active TB is polychemotherapy, which involves a combination of four drugs for at least 6 months. In the first 2 months, isoniazid, rifampicin, pyrazinamide, and ethambutol are administered, and in the following 4 months, only rifampicin and isoniazid are used to achieve pathogen elimination and prevent drug resistance [57]. Recently, it has been recognized that LTBI is not a stable condition, but rather a spectrum of infections (e.g., intermittent, transient, or progressive) that can lead to incipient, then subclinical, and finally active TB disease [58]. LTBI diagnosis is indirect and relies on detecting an immune response against MTB antigens, assuming that the immune response has developed following contact with MTB [59]. Concerning the asymptomatic condition, WHO guidelines recommend performing IGRA tests consistently in high- and medium-incidence countries and maintaining sanitary migration control toward low-incidence countries, as early identification and treatment of individuals with LTBI are a major priority for global TB control [17].

Rifamycin-based treatments have gained popularity due to their shorter treatment duration and approximately 90% efficacy in preventing progression to active TB. Similar results are obtained from isoniazid monotherapy for 6–12 months, which has been the method used for decades. However, isoniazid monotherapy presents higher risks of hepatotoxicity in comparison with rifamycin-based treatments. Due to these advantages and efficacy, the use of rifamycin in first-line treatment for LTBI is increasing [60,61]. In first-line treatment for LTBI, four main antimicrobial approaches are available: (1) isoniazid monotherapy at doses of 5–10 mg/kg up to a maximum of 300 mg/day for 6–12 months; (2) rifampicin monotherapy at a dose of 10 mg/kg/day (600 mg per day) for 3–4 months; (3) a combination of isoniazid (900 mg) with rifampicin (600 mg) for 3–4 months; and (4) a combination of isoniazid (900 mg) with rifapentine (900 mg) once a week for 3 months, more information can also be found in Table 2 [62,63].

In cases of resistant LTBI or to prevent resistance to first-line drugs, combinations of first-line antimicrobial drugs with second-line drugs have been used. Some alternatives include the combination of ethambutol (15–20 mg/kg) with first-line agents, administered once daily for 6–12 months; the combination of rifampicin (600 mg) with pyrazinamide (20–25 mg/kg) daily for 2 months; the combination of pyrazinamide with ethambutol at daily doses of 20–25 mg/kg and 15–20 mg/kg, respectively, for 9 months and monotherapy with Delamanid 100 mg twice a day [64]. Levofloxacin has proven to be effective in the treatment of drug-resistant LTBI, with doses of 10–15 mg/kg per day for up to 2 weeks. However, prolonged therapies have shown incidence of musculoskeletal complications [65]. Moxifloxacin is another antimicrobial used in the treatment of first-line drug-resistant LTBI, with daily doses of 400 to 500 mg for 6 months. It should be noted that this regimen is associated with risks of hepatotoxicity and intolerance, requiring rigorous monitoring during treatment [66,67].

**Table 2 pharmaceutics-15-02409-t002:** Drugs currently used to treat LTBI vs. active TB, targets, and recommended doses.

Treatment Line	Drug	Target	Daily Dose Active TB	Daily Dose LTBI	References
**First line**	Ethambutol	Arabinogalactan biosynthesis	15–25 mg/Kg	15–20 mg/Kg	[68,69]
	Isoniazid	Mycolic acid biosynthesis	5 mg/Kg	5–10 mg/Kg	[60,61,63,69]
	Pyrazinamide	Energy metabolism	30 mg/Kg	20–25 mg/Kg	[61,63]
	Rifampicin	RNA synthesis	10 mg/Kg	10 mg/Kg	[64,70]
	Rifapentine	RNA synthesis	900 mg/weekly	20 mg/kg	[71]
**Second line**	Delamanid	Mycolic acid biosynthesis	200 mg/day	100 mg/day	[72,73]
	Moxifloxacin	Inhibition of DNA Gyrase	7.5–10 mg/Kg	400–500 mg/day	[66,74]
	Levofloxacin	Inhibition of DNA Gyrase	15 mg/Kg	10–15 mg/Kg	[65,75]

## 5. Approaches to the Discovery of Drugs against LTBI

The WHO recognizes that the diagnosis and treatment of LTBI is an important strategy to accelerate the global reduction of TB and achieve TB eradication, as treatment can significantly reduce progression to active TB disease [76]. In 2020, the US CDC updated clinical guidelines for the treatment of LTBI, which consists of 3-, 4-, 6-, or 9-month treatment regimens, as needed [77]. However, one of the key factors limiting effective LTBI treatment is poor treatment acceptance and completion, due to the lengthy duration and significant side effects. Furthermore, another important factor for TB eradication is the increasing resistance of MTB to drugs used for treating the early stages of the disease. Thus, the development of new drugs with greater biocompatibility and efficacy is necessary to reduce treatment times and indirectly reduce the likelihood of generating new resistances.

Taking into consideration the previously mentioned information and to create new drug candidates against MTB, Moodley et al. [78] synthesized quinoline–urea–benzothiazole hybrids in modified Middlebrook 7h9 culture media, using casitone and tiloxapol (CAS) as a supplement in the first medium, and albumin–dextrose–catalase and tween 80 (ADC), the latter being a good indicator of possible protein binding. The subset of hybrids containing 1,2-propanediamine showed higher anti-MTB activity, specifically compound 6u which obtained a MIC90 of 0.968 µg/mL and 5.738 µg/mL for the CAS and ADC media, respectively. Additionally, HepG2 cells exposed to MIC90 concentrations of compound 6u obtained 100% viability.

On the other hand, Shyam et al. [79] evaluated the anti-MTB activity of a library of mycobactin analogs (siderophore). Analogs derived from pyrazoline carbothioamine showed promising activity in the MIC assay in an iron-restricted environment, achieving 90% MTB elimination at a concentration of 4 µg/mL with an IC of 32. Pyrazoline derivatives also obtained a MIC90 of 4 µg/mL under the same conditions, but it was observed that they caused self-poisoning of MTB due to the internal accumulation of siderophores. Based on these results, the authors hypothesized the existence of a mycobactin binding site in the MmpL4/MmpL5 complex of mycobacterial efflux pumps.

Ottavi et al. [80] synthesized amidinoureas to inhibit phosphopantetheinyl transferase, an enzyme involved in the biosynthesis of mycolic acids, resulting in IC50 values as low as 0.7 and MIC90 of 0.95 µM in MTB H37Rv. The compounds showed no toxicity against THP1 and HEPG2 cells, but they did exhibit signs of cardiotoxicity by inhibiting Ca and Na channels, although this inhibition was reduced in certain analogs. It has recently been reported that a concentration of 1 μM of inactive vitamin D (25(OH)D3) restored the ability of macrophages infected with MTB to express the antimicrobial peptide cathelicidin LL37, regardless of glucose concentration [81].

Basarab et al. [82] evaluated the anti-MTB activity of spiro-pyrimidinetrione analogs, all of which were found to be promising, especially analog 22, which exhibited 10 times greater activity (1.7 μM) than that of zoliflodacin and did not generate genotoxicity. In in-vivo assays in BALB/c mice, the oral administration of the compound (0.2 mL at 300 mg/kg) reduced mycobacterial load by 0.6 log10 after 12 days of treatment, and its inhibitory activity on DNA gyrase was confirmed. The decrease in anti-MTB activity was evident and was attributed to the permeability of the drug; however, in future studies, permeability issues could be resolved by using nanocarriers.

The deoxypregularin (DPG) is a compound extracted from the root of Cynanchum atratum that has shown promising results against MTB. Seo et al. synthesized analogs of DPG and obtained a MIC of up to 0.2 μg/mL. They were able to inhibit almost 100% of intracellular H37Rv and XDR MTB at 25 μg/mL, surpassing the results of isoniazid against the XDR strain and obtaining similar values in the H37Rv strain. They also infected murine models of pulmonary TB for 4 weeks at low doses and then administered 30 mg/kg of the drug for the following 4 weeks, achieving a 90% reduction in bacterial load, in comparison with the control group.

Similarly, Si et al. [83] synthesized 24 compounds derived from meroterpenoids isolated from marine sponges. All compounds showed excellent inhibition against MTB H37Ra, with values of up to 1.4 μM MIC for intracellular MTB and IC50 of 23.2 μM in J774A1 macrophages and 70.6 μM in HepG2 cells. Likewise, Qin et al. [84] developed thieno-benzenesulfonamide derivatives with inhibitory activity against the decaprenylphosphoryl-β-d-riboside 2′-epimerase enzyme, essential for the biogenesis of the Mycobacterium cell wall. The most promising compound showed a MIC of 0.24 μg/mL but resulted in an IC50 in Vero and HepG2 cells of 3.62 and 36.79 μg/mL, respectively. Additionally, Noschka et al. [85] generated an angiogenin derivative, an endogenous protein with anti-MTB activity called Angie1. This compound did not show toxicity in macrophages or zebrafish at concentrations of 27.54 and 108 μM. Furthermore, the intramacrophage activity of Angie1 loaded into liposomes was analyzed, and it was found to limit MTB multiplication from 9.5 to 5.2 times.

## 6. Promising Drug Candidates with Anti-Intracellular MTB Activity in Preclinical and Clinical Stages

Following the boom in antibiotic development, the discovery of drugs targeting TB experienced a decline. However, in the 1990s, the growing drug resistance and the surge in TB cases led the WHO to classify it as a global crisis. In 2000, representatives from various sectors (academia, pharmaceutical companies, and public–private partnerships) convened in South Africa to address the pressing need to develop new drugs and improve treatments for TB [86]. Today, after two decades of intense efforts, significant achievements have been made, with seventeen drugs at different stages of clinical development and six in preclinical phases. Out of these, approximately ten in the clinical stage could be considered to possess innovative mechanisms of action. The other seven candidates are optimized versions of already known anti-tuberculosis drugs [87]. However, only 10 of the 17 drugs in the clinical stage have been tested for their ability to eradicate intracellular MTB (Figure 3), yielding promising results and suggesting that they could be employed in the treatment of LTBI.

## 7. Candidates for Drugs with Intracellular Anti-LTBI Activity in Phase II Clinical Trials

OPC-167832: The new antitubercular drug candidate OPC-167832 was developed and commercialized by Otsuka Pharmaceutical, Inc., (Tokyo, Japan). It is a derivative of 3,4-dihydrocarbostyryl that demonstrated activity against MTB by inhibiting the DprE1 enzyme, which is involved in the synthesis of the cell wall. The MIC ranged from 0.00024 to 0.002 μg/mL [88,89]. Additionally, the MIC90 of OPC-167832 against intracellular strains of MTB H37Rv and Kurono were assessed. The results obtained for the aforementioned strains were 0.0048 μg/mL and 0.0027 μg/mL, respectively, representing a superior inhibitory activity compared with rifampicin, a reference drug used in the treatment of TB, whose MIC90 values against the same strains were 1.1501 μg/mL and 0.5648 μg/mL, respectively [89].

Sutezolid (PNU-100480): A thiomorpholine analog of linezolid but with greater activity in different experimental models of TB. Additionally, it has the potential to offer an even better safety profile. It has a MIC of 0.062 µg/mL and acts by inhibiting protein synthesis, binding to the 50S ribosomal subunit [90]. A study conducted by Zhu et al. [91] evaluated the intracellular antimicrobial activity of sutezolid against MTB, yielding a MIC value of 0.05 μg/mL. Furthermore, the cumulative intracellular activity of sutezolid was examined in TB patients receiving split doses of 600 mg twice daily and a single daily dose of 1200 mg. The split-dose regimen produced a greater cumulative effect, evidenced by a log 10 reduction per day of −0.269 (90% CI, −0.237 to −0.293) in comparison with the single daily dose of −0.186 (90% CI, −0.160 to −0.208).

BTZ-043: The University of Munich in collaboration with other scientific institutions developed BTZ-043, a Benzothiazinone that was used in phase II clinical studies in South Africa in November 2020. Its potential anti-MTB activity is due to its ability to block the activity of the enzyme DprE1, which is essential for the synthesis of D-arabinofuranose, an important component in the cell wall of MTB. This drug acts highly selectively in mycobacterial species. In addition, BTZ-043 has been shown to be effective against all evaluated MTB strains, including those isolated from MDR and XDR patients. In vitro studies have shown that the MIC ranges from ~0.1 to 80 ng/mL for fast-growing strains and between 1 and 30 ng/mL for MTB complex strains. In mouse models, BTZ-043 has been found to be more effective than INH [92,93]. Treu et al. [94] used high-resolution MALDI imaging to investigate the spatial and temporal distribution of BTZ-043 in centrally necrotizing granulomas harboring MTB. The researchers found an early accumulation of BTZ-043 in the lipid-rich foamy macrophage zone, suggesting that these immune cells act as a drug reservoir. Furthermore, they observed the penetration of BTZ-043 into the necrotic core, indicating passive diffusion in the absence of vascularization or active transport systems. The findings demonstrate that BTZ-043 is a promising clinical-stage candidate capable of efficiently reaching and penetrating TB granulomas, making it a promising drug for combatting LTBI.

Delpazolid (LCB01-0371): LegoChem BioSciences, Inc., a company based in Daejeon, Republic of Korea, has developed a new substance called Delpazolid, which is an oxazolidinone with a cyclic amidrazone that inhibits the protein synthesis of the cell wall of MTB. Studies have shown that Delpazolid is capable of reducing the minimum amount of MTB H37Rv bacteria and significantly lowering resistance rates, especially in MDR-TB strains, as compared with Linezolid. The safety, tolerability, and pharmacokinetics of Delpazolid were evaluated in a phase-1 clinical trial, and even after three weeks of repeated dosing, no adverse events, such as myelosuppression, were found. In addition, preliminary results have been reported on the efficacy and safety of the drug in a phase-2a clinical trial, which included patients with drug-sensitive TB. Phase 2b clinical trials began in October 2021 in South Africa and Tanzania [95]. Cho and Jang [96] demonstrated the ability of Delpazolid to inhibit the intracellular growth of MTB H37Rv in bone marrow-derived macrophages by measuring MIC. The results indicated that Delpazolid and Linezolid exhibit similar efficacy levels at low concentrations (1 ng/mL); however, at higher concentrations (1 µ/mL), Delpazolid showed greater efficacy, managing to eliminate almost 100% of MTB.

Telacebec (Q203): Maxwell Biotech (Boston, MA, USA) and Qurient Co., Ltd., (Seongnam, Republic of Korea) developed Telacebec, a highly effective drug in the clinical phase to treat MDR-TB. This compound is an amiloride derivative that acts as an inhibitor of both MTB ATP synthase and cytochrome bd oxidase. By blocking these enzymes, ATP synthesis is prevented, which in turn leads to the cellular death of mycobacteria. It is worth noting that, in contrast to other treatments, the efficacy of Telacebec does not depend on the replication state of the bacteria. Even Q203 showed antimicrobial activity against intramacrophage MTB H37Rv at a MIC50 of 0.28 nM [97,98,99].

SQ109: Sequella, Inc., (Rockville, MD, USA) developed SQ109, an asymmetric diamine compound derived from adamantane. This drug targets the mycobacterial membrane protein Large 3 (MmpL3), allowing interference with cell wall synthesis and bacterial activity. Its mechanism of action consists of dissipating the transmembrane electrochemical proton gradient necessary for cell wall biosynthesis, resulting in decreased bacterial viability [100,101]. SQ109 is a promising drug against LTBI. Singh et al. [101] treated MTB H37Rv-infected peritoneal macrophages with SQ109 at a concentration of 0.39 μg/mL for 24 h, successfully reducing the MTB colony-forming units to less than 10,000.

## 8. Candidates for Drugs with Intracellular Anti-tb Activity in Phase III Clinical Trials

Bedaquiline (TMC207): Diarylquinoline class drug with a unique and specific mechanism of action against the ATP synthase proton pump of MTB. This enzyme is essential in the process of energy generation in MTB. Bedaquiline has been shown to be effective against strains of MTB, MDR-TB, XDR-TB, and intermediate drug-resistant TB (IR-TB). It received conditional approval from the U.S. FDA in 2012 for the treatment of MDR-TB in combination with the current second-line treatment regimen. Its inclusion in a first-line regimen and/or as part of a more affordable and simpler regimen for drug-resistant TB could have an even greater impact on the fight against TB. Furthermore, bedaquiline has demonstrated rapid sterilizing activity, meaning that it is capable of killing TB bacteria, even in their latent phase, which is critical for the success of any anti-TB regimen [102,103,104].

Delamanid (OPC-67683): Since the early 1990s, Otsuka Pharmaceutical, Inc., has committed substantial resources towards the investigation and creation of TB remedies. This relentless pursuit has led to the development of delamanid, a top-tier bicyclic nitroimidazole, capable of hindering mycolic acid synthesis in MTB. Given its promising performance in phase IIb trials, the European Medicines Agency gave conditional approval to delamanid in 2014 for treating MDR-TB in adults. Clinical trials have substantiated both its effectiveness and safety, positioning it as an essential tool in the MDR-TB treatment arsenal [105]. Significantly, delamanid has become accessible in over 100 countries classified as low- to middle-income, with funding available through the Global Fund to Fight AIDS, TB, and Malaria. This move has provided MDR-TB patients with crucial access to this medication, marking a meaningful advancement in the battle against this ailment [106]. Liu et al. [107] investigated the efficacy of delamanid against intracellular mycobacteria using differentiated THP-1 cells, derived from human acute monocytic leukemia and infected with MTB H37Rv. The infected cells were treated with various concentrations of delamanid for 4 h. The findings of this study revealed that delamanid at a concentration of 0.1 μg/mL proved to be as effective as rifampicin at 3 μg/mL, making it a promising alternative against LTBI.

Pretomanid (PA-824): Belongs to a new class of antibacterial agents known as nitroimidazoles and presents a series of attractive characteristics as a potential therapy against TB [108]. In particular, it is highlighted for its innovative mechanism of action inhibiting mycolic acids, by the generation of nitrogen reactive species and ATP depletion, making it capable of eliminating intramacrophage MTB [109]. In addition, this compound has not shown evidence of cytotoxicity or mutagenicity in a series of clinical studies, has no significant interactions with cytochrome P450, and does not show relevant activity against a wide range of Gram-positive and Gram-negative bacteria [110]. Pretomanid is also a drug with the potential for use against LTBI and has been investigated in differentiated macrophages derived from human monocytes of the THP-1 cell line. Following a 4 h exposure period to the pharmacological agent and an additional 68 h of incubation, bacterial viability analyses revealed that the intracellular efficacy of pretomanid is comparable to that of isoniazid. However, its potency was observed to be three and two times lower than that of delamanid and rifampicin, respectively [111]. Some candidates with promising results with intracellular activity against MTB was shown in Table 3.

## 9. Are There Candidates in the Discovery of Drugs against LTBI?

Although there is no clear treatment against LTBI, some molecules have been considered promising for the treatment against MTB. The main challenge of new drugs is in the specificity and selectivity of the drug. Some highlights have been published as promising: antimicrobial peptides and drug conjugates.

Antimicrobial peptides (AMPs) are small proteins composed of less than 100 amino acids, also known as host defense peptides, naturally produced by living organisms as protection against pathogens [112,113,114]. AMPs are part of the innate immune system of most living organisms, which can alter the bacterial cell wall in various ways and then interact with enzymes, nucleic acids, and organelles [115,116,117].

AMPs have the ability to stimulate CD4+ and CD8+ cells, obstruct intercellular pathways, and increase the production of proinflammatory proteins by increasing levels of TNF-α, IL-1β, and IL-10 to eliminate intracellular bacteria. Their action is not limited to bacteria, but it also combats fungi and viruses [5,117]. Moreover, AMPs can even eliminate drug-resistant bacteria, such as methicillin/vancomycin-resistant *Streptococcus aureus*, carbapenem-resistant *Pseudomonas aeruginosa*, and MTB [118]. Therefore, AMPs are an excellent alternative to address the unmet need of finding a new therapeutic agent against TB.

Translocase I is an enzyme related to the biosynthesis of peptidoglycans, making it an unexploited but highly important target. For this reason, Tran et al. [119] synthesized analogs of Sansamicin with translocase I inhibitory capacity. To synthesize the new analogs, cyclohexylalanine and 2-naphthylalanine were added to the C-terminal end of Sansamicin, and subsequently human THP-1 macrophages were infected with MTB. Analogs containing 2-naphthylalanine eliminated intramacrophagic MTB at lower concentrations than those modified with cyclohexylalanine, with MIC50 values ranging from 0.5 to 5.1 µM, possibly due to the cell-permeability properties provided by 2-naphthylalanine.

Similarly, it was reported that the tandem repeat of a peptide derived from human chemerin reduced 80% of the CFU of *Mycobacterium smegmatis* internalized in TH1 macrophages at a concentration of 40.32 μM, without exhibiting hemolytic or cytotoxic activity [120]. In contrast, other researchers were able to synthesize a peptide from the cytolytic active site of granulysin with the ability to penetrate macrophages and eliminate MTB without causing toxicity [85]. They also reported that the antitubercular compounds produced by actinomycetes, such as ecumicin, rufomycin, and lassomycin, have the ability to inhibit the proteolytic activity of the ClpC1/ClpP1/ClpP2 complex, which in turn results in the death of the MTB bacteria.

Recently, Cotta et al. [121] evaluated the activity of anti-InhA nucleic acid peptides, which target the gene essential for the synthesis of mycolic acids in mycobacteria, at a concentration of 5 μM. They achieved a 2.4 log reduction in bacterial load of MTB H37Ra, and the activity was further enhanced by 2.8 log when the peptide was combined with 1.6 mg/L ethambutol, possibly due to the drug’s permeabilizing activity. Upon utilizing 5 μM of anti-InhA, intracellular activity proved to be less effective, achieving merely a 0.5-log reduction in bacterial load. However, when we combined 2.5 μM of anti-InhA with 0.8 mg/L of ethambutol, we observed a 1.3-fold decrease in the bacterial load. Additionally, it was reported that a synthetic peptide, IDR108 (VRLIVAVRIWRR-NH_2_), loaded onto poly(lactic-co-glycolic acid) coated with N-acetylcysteine at a concentration of 10 μg/mL, reduced the MTB UFC load internalized in RAW264.7 macrophages by 2.55 log after 48 h of incubation. The peptide showed dose-dependent toxicity, and it was found that the pure peptide was more cytotoxic than the encapsulated form, with viability of 75.69 ± 6.4% and 89.09 ± 8.82%, respectively [122].

In contrast, Upadhyay et al. [123] administered six doses of peptidic inhibitors of STAT3 at a concentration of 4 µg/kg of body weight over two weeks by the intrapulmonary route to C57BL/6 mice infected with MTB H37Rv. They achieved a 1.7- and 0.6-log reduction in UFC for the ST3-H2A2 and IL10R1-7 (Pal-KLVTLPLISSLQSSE-NH2, all-D) compounds, respectively. An increase in nitric oxide synthase, NADPH oxidase, and lysozyme activity was noted. The MIC showed that the molecules did not have anti-MTB activity, and it is believed that the reduction in bacterial load was solely due to their STAT3 inhibitory effect. Likewise, it was determined that macrocyclic peptides inhibit the MTB20S proteasome at nanomolar concentrations, with a 10,000-fold selectivity over human c-20S and i-20S. Furthermore, one of the peptides was stable in plasma, had time-dependent activity against MTB20S, and was not toxic to human cells [124].

In a similar vein, Song et al. [125] employed two methodologies to immobilize antimicrobial peptide polymers (SNAPP) onto microcapsules. The first approach involved complexing SNAPP with tannic acid (TA), while the second encapsulated it within an iron-phenolic (Fe-III-TA) coating. The loading capacity of SNAPP exhibited notable disparity based on the method implemented, reporting 0.8 pg for SNAPP-TA and 4.4 pg for SNAPP-Fe-III-TA. Interestingly, at a pH of 7.4, a sustained release with high antimicrobial efficacy was observed. Both microcapsule systems were internalized by alveolar macrophages in vitro, demonstrating the potential of polyphenol-based capsules for peptide drug immobilization and intracellular administration.

Parallelly, Ristroph et al. [126] employed an encapsulation technique to administer the AntiTB peptide ecumycin via monodisperse nanocarriers measuring 60 nm. This was achieved through a blend of techniques: FlashPrecipitation and hydrophobic ion pairing, leading to an encapsulation efficiency of 70% and a mass loading of 24%. This primary formulation was stabilized with a custom-synthesized polymer of poly(caprolactone)-block-poly(ethylene glycol)-hexamannose at three levels of mannose surface coverage (0%, 4%, and 74% of polymer chains ending in hexamannose). Upon evaluating these formulations against MTB, the encapsulated peptide displayed a reduction in CFUs by up to 3.8-log(10) units. Notably, increased antitubercular activity of ecumycin was measured from formulations prepared with larger quantities of mannose surface coverage.

There are several conjugates/analogs that are used to improve the activity against resistant strains, but we have selected only two recently published potential candidates such as sulfonamide, benzofuroxan, thiazole, triazole, thio- and semicarbazone derivatives.

Benzofuroxans belong to a chemical structure made up of a 1,2,5-oxadiazole-2-oxide, often known as furoxan. The furoxan ring is asymmetric and is distinguished by an oxygen atom made of N-oxide that is outside the ring. As a result, when R1 and R2 are different from one another, two regioisomers occur. The oxygen atom on the ring is traditionally numbered one, the nitrogen atom with the oxygen atom outside of the ring is numbered two, and the other atoms are numbered three, four, and five in that order [127]. Until now, several benzofuroxans derivatives with activity against MTB have been reported, however there are a few who presented excellent activity against MTB [128]. Fernandes and colleagues reported that the use of these derivatives would be promising candidates against LBTI since they have high specificity, even against MDR strains [129]. The reproduction and drug design techniques are described in Figure 4.

These compounds without promise likewise demonstrated that the absence of the furoxan group in the molecules would negatively affect their activity [129]. This led to the generation of a new second alternative group of molecules against MDR and XDR MTB, derivatives of Imidazo [2,1-b][1,3]oxazine, an important pharmacophore group of pretomanid and delamanid (Figure 5). In general, these compounds did not present cytotoxicity and they added that the possible mechanism of action would be related to the interaction with the deazaflavin-dependent nitroreductase holoenzyme present in pretomanid with MIC values < 0.5 μM, which proves their potential against LTBI since they are capable of crossing macrophages and entering the bacterial membrane of dormant strains [130].

On the other hand, sulfonamide derivatives have also generated great importance against LTBI since their analogues have not shown cytotoxicity and various microbiological applications have been reported. Its formulation is shown in Figure 6 and the obtaining of the various analogues with potential activity against MTB is explained; likewise, the authors showed their potential against LTB strains with high selectivity indices.

## 10. Conclusions and Expert Opinion

TB, and specifically LTBI, remains one of the primary challenges in global health, requiring reliable and early detection methods. Emerging diagnostic tools, such as pulmonary immunity, epigenetics, metabolomics, and TBST, are transforming the diagnostic field, providing faster, more accurate, and more sensitive results compared with conventional techniques. Such advancements have the potential to significantly enhance the LTBI detection and monitoring process. On the other hand, the drive to find more effective treatments for LTBI has led to the identification of therapeutic agents with great potential, many of which are already in preclinical and clinical evaluation stages. The role of AMPs is particularly noteworthy for their ability to alter bacterial cell membranes and tackle resistant pathogens. Studies focused on enzymes such as Translocase I, and particularly benzofuroxan derivatives have proven highly effective against various TB strains. Despite significant progress, we still face substantial challenges. It is essential to persist in research to overcome current barriers and refine both diagnostic and therapeutic strategies. The implications of these advancements hold transformative potential, paving the way for controlling and possibly eradicating LTBI on a global scale.

## Figures and Tables

**Figure 1 pharmaceutics-15-02409-f001:**
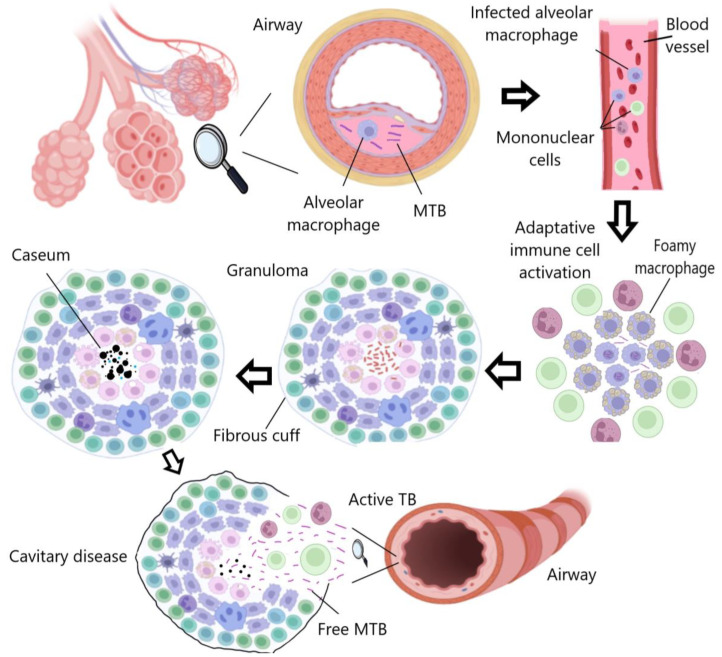
TB progression: Macrophages attempt to eradicate MTB, which persists intracellularly and induces an inflammatory response, leading to granuloma formation. The infection may become latent or progressive, resulting in cavity rupture and bacterial dissemination. Created with Biorender.

**Figure 2 pharmaceutics-15-02409-f002:**
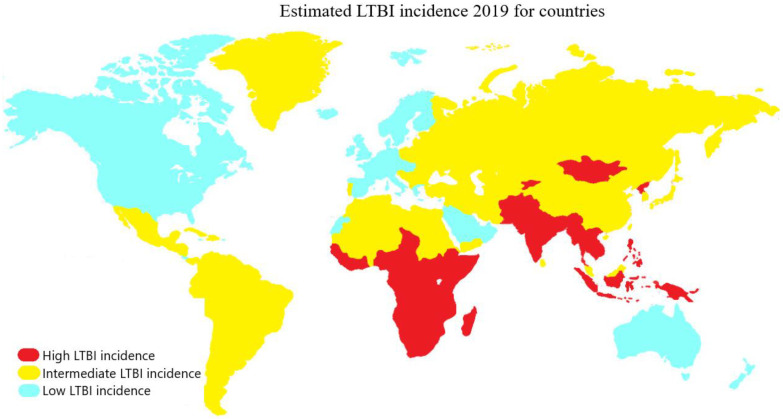
Countries with LTBI incidence are classified according to their prevalence into high (red), intermediate (yellow), and low (light blue) categories, corresponding to prevalence ranges of 28–36%, 19–20%, and 3–5%, respectively.

**Figure 3 pharmaceutics-15-02409-f003:**
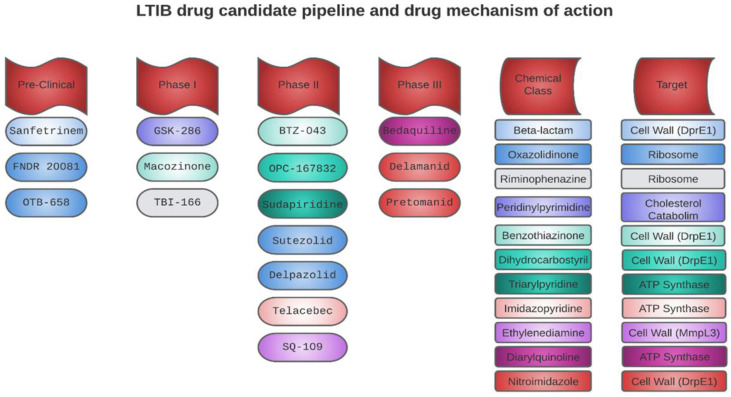
Drug candidates with potential activity against LTBI in preclinical and clinical stages, their mechanisms of action, and chemical classes.

**Figure 4 pharmaceutics-15-02409-f004:**
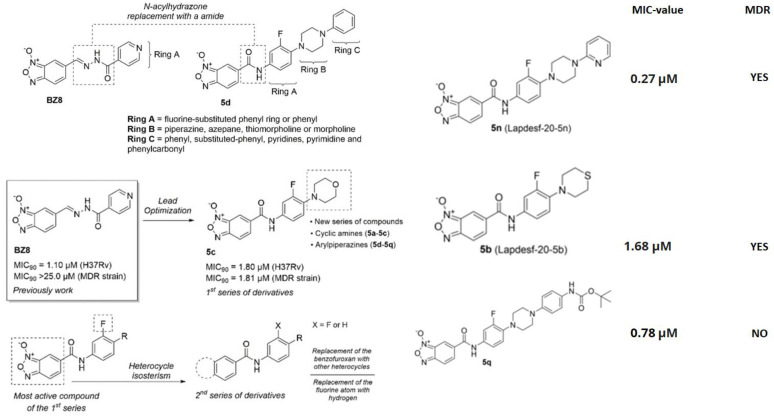
Scheme of the design of drugs based on benzofuroxan derivatives, and an example of the expansion of the chain of rings linked to the benzofuroxan nucleus. Reprinted from Fernandes et al. [129], with permission from the John Wiley and Sons License 2023.

**Figure 5 pharmaceutics-15-02409-f005:**
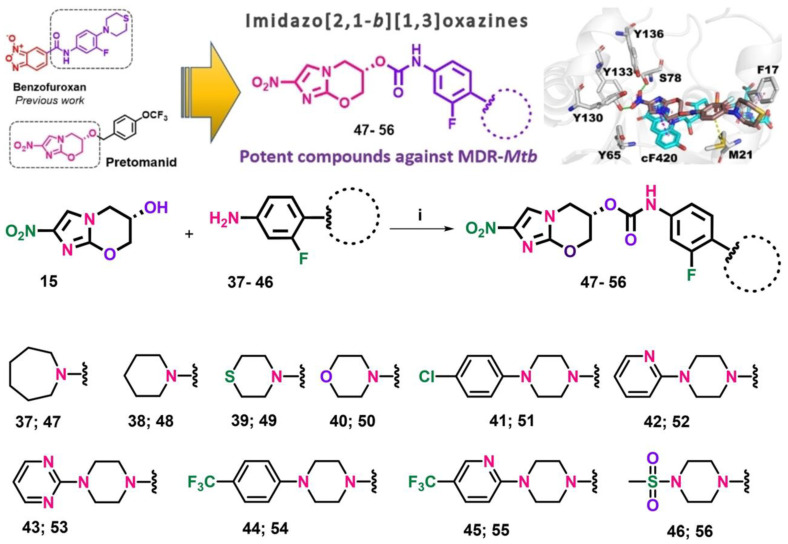
Synthetic route for the preparation of final carbamate compounds **47**–**56**. Reagents and conditions: (i) triphosgene, triethylamine, THF, 20 °C, 16 h. Reprinted from Fernandes et al. [130], with permission from the John Wiley and Sons License 2023.

**Figure 6 pharmaceutics-15-02409-f006:**
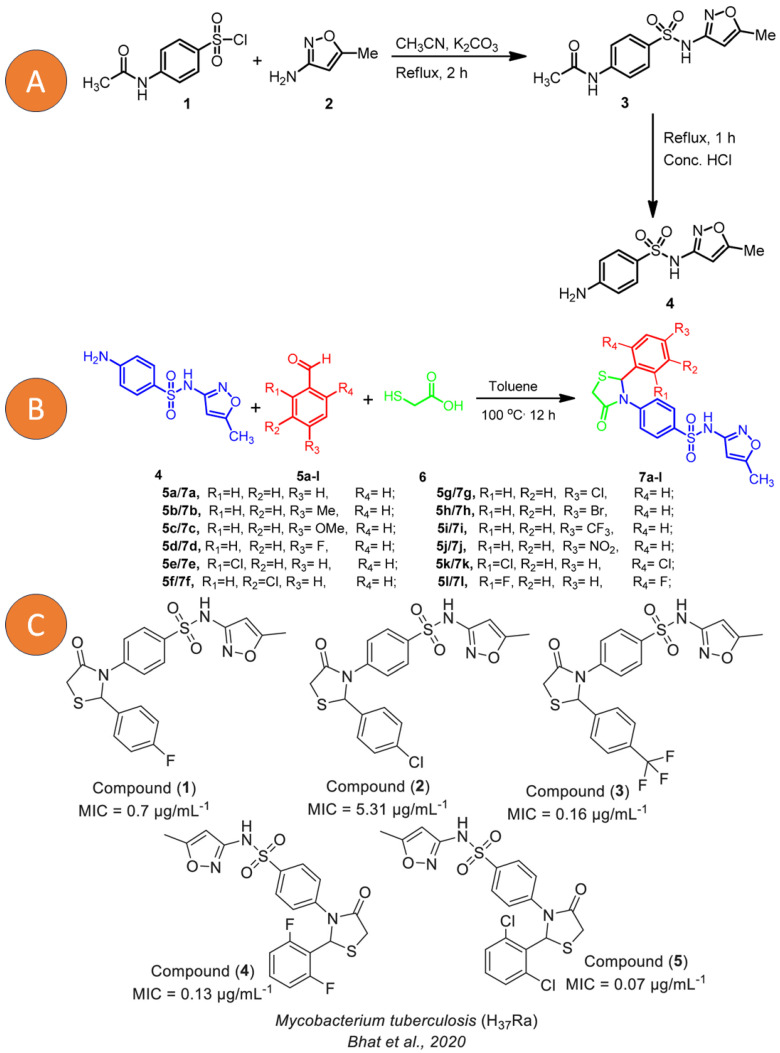
(**A**) Synthesis of 4-amino-N-(5-methylisoxazol-3-yl)benzenesulfonamide (**4**). (**B**) Synthesis of sulfamethaoxazole 4-thiazolidinone hybrids (**7a**–**l**). (**C**) New Sulfamethaoxazole-based 4-thiazolidinone derivatives (**1**–**5**). Reprinted from Bhat et al. [131] with permission from the MDPI open access license 2020 and Scarim et al. [132] with permission from John Wiley and Sons License 2023.

**Table 1 pharmaceutics-15-02409-t001:** New diagnostic methods for the detection of TB. Data obtained from: The World Health Organization consolidated guidelines on tuberculosis: module 3: diagnosis: rapid diagnostics for tuberculosis detection, 2021 update.

Test	Purpose	Application	Population	Principle
Xpert MTB/RIF	Detect MTB and RIF resistance	Pulmonary TB, extra-pulmonary TB, HIV co-infection	Adults and children	PCR
Xpert MTB/RIF Ultra	Detect MTB, minimize false RIF resistance results	TB meningitis, pulmonary TB, extra-pulmonary TB	Adults and children	PCR
Truenat MTB, MTB Plus, and MTB RIF Dx tests	Semi-quantitative detection of MTB complex, RIF resistance	Pulmonary TB, HIV co-infection	Adults	PCR
TB-mediated isothermal DNA amplification (LAMP)	Detect MTB	Pulmonary TB	Adults	PCR
Loop-LAMP	Detect mutations associated with drug resistance to RIF, INH, and ETO	Pulmonary TB, extra-pulmonary TB	Adults	PCR
Lipoarabinomannan (LAM) determination by lateral flow immunochromatography	Detect mycobacterial LAM antigen in urine	Pulmonary TB, extra-pulmonary TB, HIV co-infection	Adults and children	Immunochromatograph

**Table 3 pharmaceutics-15-02409-t003:** Second- and third-phase pharmacological agents with intracellular activity against MTB.

Drug	LTBI Activity	References
OPC-167832	The intracellular activity recorded against the H37Rv strain was 0.0048 μg/mL, and 0.0027 μg/mL for the Kurono strain, demonstrating a superior efficacy compared with RIF	[88,89]
Sutezolid	It boasts a safety profile that surpasses that of linezolid, with an intracellular MIC of 0.05 μg/mL, and additionally exhibits a pronounced cumulative effect.	[91]
BTZ-043	It has shown efficacy against MTB MDR and XDR strains at concentrations ranging from 1 to 30 ng/mL, uniquely accumulating in foamy macrophages and adeptly penetrating necrotic nuclei.	[92,93,94]
Delpazolid	It manifests intracellular activity in bone marrow-derived macrophages. Its anti-MTB potency parallels that of Linezolid at 1 ng, yet it surpasses Linezolid’s efficacy at a concentration of 1 μg.	[96]
Telacebec	Telacebec’s antimycobacterial activity remains consistent regardless of the mycobacterial replication state. Notably, exhibited intramacrophagic antimicrobial activity against MTB H37Rv with an MIC50 of 0.28 nM.	[97,98,99]
SQ109	In studies involving peritoneal macrophages infected with MTB H37Rv, SQ109 at a concentration of 0.39 μg/mL over 24 h significantly reduced MTB colony-forming units to less than 10,000.	[101]
Bedaquiline	It demonstrates an eminent ability to eradicate MTB, MDR-TB, XDR-TB, and IR-TB. Its swift sterilizing action also affords it activity against LTBI.	[102,103,104]
Delamanid	In assays with THP-1 cells, it exhibited activity against intracellular mycobacteria at a concentration of 0.1 μg/mL after 4 h, positioning it as 30 times more potent than RIF in the treatment of LTBI.	[107]
Pretomanid	Following a 4 h exposure of THP-1 macrophages infected with MTB to pretomanid, there was an effective eradication of LTBI at concentrations akin to those of INH.	[111]

## Data Availability

Not applicable.

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
