# Peer review of "Advances in Diagnostics and Drug Discovery against Resistant and Latent Tuberculosis Infection"

_pharmaceutics, 2023, doi:10.3390/pharmaceutics15102409_

Round 1

Reviewer 1 Report

In the article colleagues analyzed methods for diagnosing latent tuberculosis infection, which is relevant.

The following items should be corrected:

-          The structure of the abstract should be corrected and should include: background; objective; results; limitations, and conclusion of the study

-          The Keywords should be corrected and presented more than five actual words.

-          Reference  In the introduction in  line 84  is not clear (WOS , 2022);

-            TB epidemiology data based on the WHO Reports 2021 – 2022 should be included in the introduction. 

-          In the introduction and in the methods of the study the definitions of LTBI should be applied from WHO guidelines:

- Latent tuberculosis infection: updated and consolidated guidelines for programmatic management. Geneva: World Health Organization; 2018 (https://apps.who.int/iris/handle/10665/260233, accessed 23 February 2022).

- WHO consolidated guidelines on tuberculosis. Module 5: management of tuberculosis in children and adolescents. Geneva: World Health Organization; 2022.

-          The presented data have some historical interest but currently we have new immunological methods that presented in the WHO guideline 2022 (WHO Consolidated Guidelines on Tuberculosis. Module 5: Management of Tuberculosis in Children and Adolescents; World Health Organization: Geneva, Switzerland, 2022).  New methods of immunodiagnostics have been introduced in Russia, in China others countries (such as testing with a recombinant tuberculosis allergen (Diaskintest®)), ELISPOT, QuantiFERON-TB Gold).

-          Colleagues can find necessary information in this article:  Starshinova, A.A.; Dovgalyk, I.; Malkova, A.M.; Zinchenko, Y.S.; Pavlova, M.V.; Belyaeva, E.; Basantsova, N.Y.; Nazarenko, M.; Kudlai, D.A.; Yablonskiy, P. Recombinant tuberculosis allergen (Diaskintest®) in tuberculosis diagnostic in Russia (meta-analysis). Int. J. Mycobacteriology 2020, 9, 335–346.

-          The aim of the study should be presented in the end of the introduction.

-          It should be noticed that the results in these studie can be relevant for the authors:  Park, C.H.; Park, J.H.; Jung, Y.S. Impact of Immunosuppressive Therapy on the Performance of Latent Tuberculosis Screening Tests in  Patients with Inflammatory Bowel  Disease: A Systematic Review and Meta-Analysis. J. Pers. Med.. 2022 Mar 21;12(3):507. doi: 10.3390/jpm12030507.

-          Conclusions are not clear (please look also at the first comment regarding the aim of the study).

Author Response

REVIEWER #1:

The following items should be corrected:

-          The structure of the abstract should be corrected and should include: background; objective; results; limitations, and conclusion of the study

Full agreement is found with the reviewer's feedback, and the suggested amendment has been applied to enhance the clarity of the text.

-          The Keywords should be corrected and presented more than five actual words.                                                                                                  

This insight is appreciated. Previously, we were unaware of the minimum keyword requirement, but the necessary adjustment has now been made.

-          Reference In the introduction in line 84 is not clear (WOS, 2022);

An oversight occurred regarding the citation. Initially, the paragraph referenced the World Health Organization in the American Psychological Association 7th edition style. This has been rectified with an appropriate rephrasing.

-            TB epidemiology data based on the WHO Reports 2021 – 2022 should be included in the introduction. 

The observation is highly valued. Adopting the insightful suggestion will provide additional context, leading to the modification of the text.

-          In the introduction and in the methods of the study the definitions of LTBI should be applied from WHO guidelines: Latent tuberculosis infection: updated and consolidated guidelines for programmatic management. Geneva: World Health Organization; 2018 (https://apps.who.int/iris/handle/10665/260233, accessed 23 February 2022). WHO consolidated guidelines on tuberculosis. Module 5: management of tuberculosis in children and adolescents. Geneva: World Health Organization; 2022.

Alignment with the reviewer's feedback prompted the edit to the text, aiming to improve the article's comprehensibility.

-          The presented data have some historical interest but currently we have new immunological methods that presented in the WHO guideline 2022 (WHO Consolidated Guidelines on Tuberculosis. Module 5: Management of Tuberculosis in Children and Adolescents; World Health Organization: Geneva, Switzerland, 2022).  New methods of immunodiagnostics have been introduced in Russia, in China others countries (such as testing with a recombinant tuberculosis allergen (Diaskintest®)), ELISPOT, QuantiFERON-TB Gold).

Acknowledging the reviewer's point, certain new diagnostic methods introduced in the consolidated WHO guidelines, such as the recombinant Diaskintest, were not previously covered. However, in the "New Advances in Latent TB Diagnosis" section, methods like ELISPOT, QuantiFERON-TB Gold, Genepop, and LIOSpot TB, as well as metabolomics-based methods, have been extensively discussed. The aforementioned method has been incorporated to provide comprehensive information

-          Colleagues can find necessary information in this article:  Starshinova, A.A.; Dovgalyk, I.; Malkova, A.M.; Zinchenko, Y.S.; Pavlova, M.V.; Belyaeva, E.; Basantsova, N.Y.; Nazarenko, M.; Kudlai, D.A.; Yablonskiy, P. Recombinant tuberculosis allergen (Diaskintest®) in tuberculosis diagnostic in Russia (meta-analysis). Int. J. Mycobacteriology 2020, 9, 335–346.

The information was added

-          The aim of the study should be presented in the end of the introduction.

We appreciate the recommendation. The objective was added

-          It should be noticed that the results in these studie can be relevant for the authors:  Park, C.H.; Park, J.H.; Jung, Y.S. Impact of Immunosuppressive Therapy on the Performance of Latent Tuberculosis Screening Tests in  Patients with Inflammatory Bowel  Disease: A Systematic Review and Meta-Analysis. J. Pers. Med.. 2022 Mar 21;12(3):507. doi: 10.3390/jpm12030507.

 We appreciate the recommendation. The reference was added

-          Conclusions are not clear (please look also at the first comment regarding the aim of the study).

Conclusions underwent a rewrite to reflect the feedback.

Reviewer 2 Report

The manuscript titled "Advances in Diagnostics and Drug Discovery Against Resistant and Latent Tuberculosis Infection" focuses on latent tuberculosis infection. The authors introduce drugs that can be notable in terms of sensitivity, which could potentially become part of new therapeutic regimens for multidrug-resistant tuberculosis control.

Overall, the figures and tables are generally suitable for readers, and the paragraph arrangement, reference citation, and the quality of content descriptions are excellent. Therefore, I recommend acceptance after addressing minor points.

Minor points

1. Abstract

Spelling error.

On line 28, "n" should be replaced with "In."

2.Introduction

The text within Figure 1 in the Introduction is blurry. If it's a referenced figure, please add the reference, and if it's an original figure, consider using clearer colors for improved visibility.

3. Results Section

Across named paragraphs 2-9, the explanations are concise and well-structured. However, it would be even more helpful if there were a summarized table that included drugs and therapies for quick reference.

4.Conclusion

Expert opinion or evidence is needed. If it's the author's perspective, consider simply naming the chapter "Conclusion" instead of "Expert Opinion."

Author Response

REVIEWER #2:

- Abstract

Spelling error.

On line 28, "n" should be replaced with "In."

We appreciate your attention to detail and have corrected the mentioned error.

 -Introduction

The text within Figure 1 in the Introduction is blurry. If it's a referenced figure, please add the reference, and if it's an original figure, consider using clearer colors for improved visibility.

 Noted regarding the image. Given its original nature, we've enhanced its quality.

  1. Results Section

Across named paragraphs 2-9, the explanations are concise and well-structured. However, it would be even more helpful if there were a summarized table that included drugs and therapies for quick reference.

 In response to your suggestion, we've incorporated Table 1 and 3, summarizing the diagnostic methods and drugs against LTBI in the respective phases.

4.Conclusion

Expert opinion or evidence is needed. If it's the author's perspective, consider simply naming the chapter "Conclusion" instead of "Expert Opinion."

Your feedback is valued, and the suggested change has been implemented.

Reviewer 3 Report

Both diagnosis and treatment of latent tuberculosis infection (LTBI) are an essential measure for tuberculosis (TB) control worldwide. These strategies are limited, however, by the lack of predictive ability of current diagnostic tests and the long duration and toxicity of treatment regimens. Interferon-gamma release assays (IGRAs) are more specific and sensitive than the tuberculin test and therefore allow better selection of cases requiring treatment. However, their ability to predict the development of active TB remains poor. In addition, therapeutic regimens against LTBI are long and adherence rates are low. This review discusses the use of available diagnostic techniques and new approaches to the diagnosis and treatment of ITBL, highlighting the need to develop new molecules with high therapeutic activity against sensitive and multi-drug-resistant Mycobacterium tuberculosis.

The review is in line with the objectives set out in the Special Issue "Bioactive Agents for the Treatment against Tuberculosis" and I consider that the paper can be published in the current format.

Author Response

The authors appreciate your valuable feedback.